# Preparation and Evaluation of Peptides with Potential Antioxidant Activity by Microwave Assisted Enzymatic Hydrolysis of Collagen from Sea Cucumber *Acaudina Molpadioides* Obtained from Zhejiang Province in China

**DOI:** 10.3390/md17030169

**Published:** 2019-03-15

**Authors:** Huo-Xi Jin, Hong-Ping Xu, Yan Li, Qian-Wei Zhang, Hui Xie

**Affiliations:** Zhejiang Provincial Engineering Technology Research Center of Marine Biomedical Products, School of Food and Pharmacy, Zhejiang Ocean University, Zhoushan 316022, China; xhp3201@sina.com (H.-P.X.); m18625623399@163.com (Y.L.); zjouzqw@163.com (Q.-W.Z.); 18158050952@163.com (H.X.)

**Keywords:** *Acaudina molpadioides*, antioxidant peptides, collagen, microwave-assisted

## Abstract

The present study was focused on the preparation and characterization of the antioxidant peptides by microwave-assisted enzymatic hydrolysis of collagen from sea cucumber *Acaudina molpadioides* (ASC-Am) obtained from Zhejiang Province in China. The results exhibited the effects of microwave irradiation on hydrolysis of ASC-Am with different protease. Neutrase was selected from the four common proteases (papain, pepsin, trypsin, and neutrase) based on the highest content and DPPH scavenging activity of hydrolysate Fa (Molecular weight < 1 kDa). The content and 2,2-diphenyl-1-picrylhydrazyl (DPPH) scavenging activity of Fa obtained by hydrolysis of neutrase increased by 100% and 109% respectively at a microwave power of 300 W compared with no microwave irradiation. Five subfractions were obtained after performing the gel filtration chromatography, and the Fa.2 exhibited the highest DPPH scavenging activity. The amino acid analysis showed that the contents of Glutamic acid, Alanine, Tyrosine, and Phenylalanine in fraction Fa.2 increased significantly, but an obvious decrease in the content of Glycine was observed compared to Fa. Four peptides (Fa.2-A, Fa.2-B, Fa.2-C, and Fa.2-D) were purified from Fa.2 by high performance liquid chromatography, and Fa.2-C showed the highest DPPH scavenging activity. The sequence of Fa.2-C was identified as Phenylalanine-Leucine- Alanine-Proline with a half elimination ratio (EC_50_) of 0.385 mg/mL. The antioxidant activity of Fa.2-C was probably attributed to the small molecular sizes and the presence of hydrophobic amino acid residues in its sequence. This report provided a promising method for the preparation of antioxidant peptides from collagen for food and medicinal purposes.

## 1. Introduction

Collagen, an important structural protein, constitutes about 30% of the total protein in the multicellular organisms [1,2]. Collagen is the main component of bone, fascia, tendons, cartilage, skin, and other connective tissues [3,4]. It is widely used in the fields of food and medicine due to its unique structural and biological characteristics, such as high tensile strength, low antigenicity, good biocompatibility and biological activity [5,6,7]. However, in vivo absorption of collagen is compromised due to its large molecular weight. Thus, in the last decade, collagen peptides have received more and more attention, and are considered as important components based on their excellent penetrability and bioactivity [8,9,10,11,12]. Recently, many reports suggest that collagen peptides (GFRGTIGLVG, GPAGPAG, GFPSG, and TAGHPGTH) contain diverse bioactivities, including antimicrobial, antioxidant, and antitumor activities [13,14,15].

In the human body, an imbalance between free radicals and antioxidants causes various diseases, such as skin damage, cancer, and neurological disorders [16,17]. Antioxidants can protect the body from the harmful effects of different free radicals such as superoxide, hydroxyl, and hydrogen peroxide. So, peptides with antioxidant activities might have potential applications in the pharmaceutical, medicinal, and nutraceutical fields. Bioactive peptides commonly contain 3-20 amino acids, and their antioxidant activities depends upon their amino acid compositions and sequences [18]. Amino acids, such as Proline (Pro), Glutamic acid (Glu), Tyrosine (Tyr), Phenylalanine (Phe), Arginine (Arg), and Histidine (His), can improve the antioxidant activity of the bioactive peptides [19]. Collagen contains high amounts of Pro, which protects cells from the oxidative damages induced by the free radicals. Therefore, it is very promising to produce natural antioxidant peptides from collagen.

Collagen peptides can be obtained from either acidic or enzymatic hydrolysis of collagens, but enzymatic hydrolysis is generally preferred, as it is safer, cheaper, more moderate, and less destructive than acid hydrolysis [20]. Unfortunately, the conventional enzymatic hydrolysis is time consuming as the triple-helical structure of collagen diminishes the degree of hydrolysis. Microwave radiation can mobilize the triple-helical part and reach into the inner structure of collagen. Therefore, microwave-assisted enzymatic hydrolysis of collagens can drastically reduce the reaction time from a few hours to several minutes and improve the degree of hydrolysis. Lin et al. reported that microwave assisted hydrolysis was a fast and reliable method for preparing antioxidant peptides from bone collagen [4]. Zhang et al. suggested that both the yields of the peptides and degree of hydrolysis were higher in microwave coupled enzymatic digestion than those in enzymatic hydrolysis alone [20]. Thus, microwave-assisted enzymolysis is a promising method for synthesizing bioactive peptides.

Sea cucumbers, of which more than 1400 species exist around the world, are important marine resources containing high amounts of collagen in their body wall [21]. As a traditional sea cucumber, *Acaudina molpadioides* live in the soft mud bottom from the intertidal zone to the water depth of 80 meters, and are distributed in coastal waters such as Guangxi, Guangdong, Fujian, Zhejiang, and Jiangsu provinces. *Acaudina molpadioides* contains many bioactive substances, such as collagen, saponin, and polysaccharide, though this species is still very cheap due to the lack of any highly value-added application. Recently, collagens have been prepared from many sea cucumbers, as well as characterized, and used for the preparation of bioactive peptides [1,22,23,24,25,26,27]. However, little information is available about the enzymatic preparation of bioactive peptides from the collagen of *Acaudina molpadioides* (ASC-Am). In this study, antioxidant peptides were prepared for the first time using microwave assisted enzymatic hydrolysis of the collagen from the body wall of *Acaudina molpadioides.* Furthermore, a novel antioxidant peptide was also isolated from the hydrolysate of collagen, and its amino acid sequence and antioxidant activity against 2,2-diphenyl-1- picrylhydrazyl (DPPH) radicals were determined.

## 2. Results and Discussion

### 2.1. Effect of Microwave Power on Enzymatic Hydrolysis of ASC-Am

Traditionally, collagen is hydrolyzed by acidic and enzymatic methods. However, the degree of hydrolysis of collagen is incomplete, and the molecular distribution of hydrolysate is uneven in the traditional methods. Microwave digestion is an effective method for extracting and hydrolyzing collagen [28,29]. The energy of microwaves can penetrate to the interior of collagen molecule and prevent its aggregation, resulting in complete hydrolysis. At present, the antioxidant peptides derived from protein hydrolysate have attracted broad attention. Radical scavenging assays are widely used for predicting the antioxidant activities of protein hydrolysates and peptides. Here, the DPPH method was selected for evaluating the antioxidant activity of the hydrolysate of ASC-Am. In addition, many studies have shown that the antioxidant activity of protein hydrolysate depends on its molecular weight (Mw) and amino acid composition [15,18]. Thus, three fractions with different Mw from hydrolysates of ASC-Am, including Fraction a (Fa, Mw < 1 kDa), Fraction b (Fb, 1 kDa < Mw < 5 kDa), and Fraction c (Fc, Mw > 5 kDa), were obtained by hydrolyzing with four proteases, including papain, pepsin, trypsin and neutrase, and the effect of microwave power on their contents and DPPH scavenging activities were determined.

Figure 1a shows that the contents of Fb and Fc declined by papain with increasing the power of the microwave radiation in the range of 0–150 W, resulting in an increase of Fa contents. However, as the power of the microwave radiation was increased from 150 to 300 W, the content of Fc did not change significantly, while the content of Fb was increased, which in turn reduced the Fa content. These results indicated that high power of microwave radiation was not conducive for hydrolyzing Fb with papain into the smaller molecular weight fraction Fa. According to Figure 1b, fraction Fc exhibited the highest DPPH scavenging activity among these three fractions. At higher power (100 to 300 W), the microwave radiation diminished the DPPH scavenging activities of Fc. However, the DPPH scavenging activities of Fa and Fb improved while enhancing the microwave power from 100 to 250 W.

The effect of microwave radiation on the pepsin-induced hydrolysis of ASC-Am was also investigated, and the results are shown in Figure 2. As shown in Figure 2a, the fraction Fa revealed the most content in hydrolysate of ASC-Am by pepsin catalysis, which increased from 41% (without microwave irradiation) to 64% (300 W of microwave power). Correspondingly, the contents of Fb and Fc decreased with increasing of microwave power. Pepsin generally cleaves the peptide bonds linked with either an aromatic or an acidic amino acid. Microwaves can expose the inner amino acids of the peptides to pepsin. Microwave radiation significantly improved the contents of the low molecular weight peptides (Mw < 1 kDa), indicating that the aromatic or acidic amino acids in ASC-Am might be located at the inner side of the molecule. Among the three fractions, the highest DPPH scavenging rate was observed in Fc. However, the antioxidant effect increased at first and then decreased when the power of the microwave irradiation was increased. The fraction Fa was obtained from pepsin hydrolysis without exposing to the microwave, which showed very low DPPH scavenging rate (1.2%); however, the same fraction, when irradiated with microwaves (300 W), exhibited a high DPPH scavenging rate (16.1%). Thus, the microwave irradiation increases the hydrolysis of ASC-Am by pepsin and produces many small molecule antioxidant peptides.

According to Figure 3a, the effects of microwave radiation on the content of the peptides were not as significant after trypsin hydrolysis as those after pepsin or papain hydrolysis. The content of Fc decreased slightly with increasing of microwave power (*p* < 0.05), resulting in increasing slightly of the content of Fb (*p* < 0.05), while the content of Fa remained relatively stable. The main active sites of trypsin are the carboxyl groups of lysine and arginine residues. Under microwave-assisted trypsin-induced hydrolysis, the content of small peptide Fa was found to be lower than those by papain or pepsin, indicating that the contents of lysine and arginine residues might be lower in ASC-Am. In addition, Fa remained stable while increasing the power of the microwave irradiation, which suggested that lysine and arginine residues might be evenly distributed in the collagen molecule. However, as shown in Figure 3b, the microwave power had an obvious effect on the antioxidant activity of both Fb and Fc. When the microwave power was 250 W, the DPPH scavenging rate of Fb reached the highest 26%, but the lowest 10.8% for Fc. In addition, Fa obtained by trypsin hydrolysis displayed very low DPPH scavenging rate. 

Figure 4a suggests that the contents of peptides under microwave-assistance neutrase hydrolysis was similar to that obtained from pepsin. As the power of microwave irradiation increased, the content of Fa increased, but the contents of Fb and Fc decreased. The content of Fa was approximately 78% at 300 W, suggesting that large amounts of small molecular peptides with 2–6 amino acid residues were present in the hydrolysate. Neutrase is a sequence-unspecific exopeptidase which can hydrolyze macromolecular proteins into small peptides or amino acids due to its high degree of hydrolysis. Thus, neutrase is widely used in the research and development of aquatic extracts, peptones and peptides [4]. Lin et al. showed that approximately 82% of Mw distributions were found in the range of <1 kDa in the neutrase-induced hydrolysis of bone collagen after microwave irradiation [4]. The antioxidant activity of the hydrolysate is associated with the molecular weight. Generally, low molecular weight peptides show high antioxidant activity. According to Figure 4b, the lower molecular weight fraction Fa exhibited the highest DPPH scavenging rate among the three fractions. Similar results were reported in other studies. For example, Jang et al. reported that peptides (<3 kDa) from sandfish protein hydrolysates possessed the strongest DPPH radical scavenging activity [30]. Chi et al. reported that the fraction BSH-III (Mw < 1 kDa) exhibited higher antioxidant activity than the fraction BSH-I (Mw > 5 kDa) and BSH-II (1 ≤ Mw ≤ 5 kDa) from protein hydrolysates of bluefin leatherjacket skin [31]. In addition, Figure 4b showed that microwave irradiation could significantly promote the DPPH radical scavenging activity of fraction Fa but Fb and Fc from hydrolysates by neutrase. The DPPH radical scavenging activity of Fa increased from 18.0% without being exposed to microwave irradiation to 37.7% at 150 W. A plateau in the antioxidant activity of Fa was observed at a microwave power > 150 W. This result may be attributed to the fact that an increase in microwave power does not significantly alter the composition of the amino acids of Fa.

Apart from molecular weight, the antioxidant activity of hydrolysate depends upon the amino acid composition. Different proteases have different cleavage sites, producing different peptides with the unique amino acid sequences. Thus, Fa from the hydrolysate of ASC-Am after neutrase-induced hydrolysis showed higher antioxidant activity than Fa from the hydrolysate of ASC-Am after trypsin-induced hydrolysis. In addition, the distribution of the cleavage sites in the structure of ASC-Am was different, leading to different effects of microwave irradiation on the hydrolysis with different proteases. However, the amino acid composition of collagen in Acaudina molpadioides from different sea regions may vary, so the results were restricted to the specific samples from Zhejiang province in China. Finally, Fa from the hydrolysate of ASC-Am after microwave assisted neutrase-induced hydrolysis showed the highest content and DPPH scavenging activity among all the fractions obtained after papain, pepsin, trypsin, and neutrase-induced hydrolysis.

### 2.2. Effect of Time of Microwave Irradiation on the Neutrase-Induced Hydrolysis of ASC-Am

Microwave radiation can penetrate into the interior of proteins, loosen their spatial structures and expose the reaction sites of protease [32]. The exposure of the reaction sites can accelerate the enzymatic digestion process [33]. Thus, the microwave irradiation can shorten the reaction time more effectively than the enzymatic hydrolysis alone. The effect of the duration of microwave irradiation on the neutrase-induced hydrolysis of ASC-Am was investigated. The reaction conditions were maintained as the ASC-Am concentration of 1%, neutrase dosage of 5000 U/g, temperature of 55 °C, pH 7.0, and microwave power of 300 W. Figure 5a shows that with increasing the incubation time, the content of Fa increased, but the contents of Fb and Fc decreased. After 30 min, the contents of Fa, Fb, and Fc showed no obvious change. However, the DPPH scavenging activity of Fa dramatically was changed with the incubation time. The DPPH scavenging activity of Fa was increased during 5-30 min, but decreased significantly with prolongation of the incubation time. This result may be attributed to the facts that: (1) microwave radiation over a long time period affected the physiological activity [34]. For example, the antioxidant peptides reacted with environmental oxygen [20]; (2) some peptides in the fraction Fa were further hydrolyzed to amino acids with prolonging of incubation time, resulting in a decrease in the antioxidant activity.

### 2.3. Antioxidant Activity of Fa 

According to the results, the fraction Fa from the hydrolysate after microwave-coupled neutrase induced hydrolysis of ASC-Am exhibited the highest content and DPPH scavenging rate. In this section, both DPPH and ABTS scavenging assays were performed to evaluate the antioxidant activity of Fa at different concentrations and times. As shown in Figure 6, Fa obtained after microwave induced neutrase hydrolysis of ASC-Am could scavenge DPPH and ABTS radicals in a dose-dependent manner. At the concentrations of Fa below 4 mg/mL, the ABTS radical scavenging activity was higher than DPPH radical scavenging activity. The ABTS radical scavenging rate reached 80% at 2 mg/mL concentration of Fa, while the same scavenging rate of DPPH radical was obtained at 4 mg/mL concentration of Fa. In a different study, the protein hydrolysate fraction (Mw < 1 kDa) from bluefin leatherjacket skin showed DPPH scavenging activity with the half elimination ratio (EC_50_) values of 3.574 mg/mL [31]. Pan et al. reported that the protein hydrolysate fraction SCH-I (Mw < 1 kDa) from skate cartilage revealed DPPH scavenging activity with the EC_50_ values of 11.86 mg/mL [35]. The EC_50_ values of Fa for DPPH scavenging activity was 1.31 mg/mL (data not shown), which was significantly higher than those reported earlier. 

### 2.4. Gel Filtration Chromatography of Fa

Gel filtration is an effective method for isolating the peptides from the protein hydrolysates based on their molecular size. Generally, the peptides with higher molecular size are preferentially eluted. In order to further prepare and characterize the antioxidant peptides from the hydrolysates after microwave-assisted neutrase-induced hydrolysis of ASC-Am, sample Fa was subsequently isolated into five subfractions (Fa.1, Fa.2, Fa.3, Fa.4, and Fa.5) on a Sephadex G-15 column (Figure 7a). The subfraction Fa.2 possessed the highest concentration in the fraction Fa. Furthermore, the DPPH scavenging activity of each subfraction was determined at the concentration of 0.2 mg/mL. Figure 7b suggests that Fa.2 possessed the highest DPPH scavenging activity among the five subfractions, followed by Fa.1, Fa.3, and Fa.5, whereas Fa.4 showed the lowest activity. Actually, Fa.2 presented 25.8% of DPPH scavenging activity at 0.2 mg/mL, which is very low compared to that of Vc at the same concentration. Based on this result, further studies would focus on the separation of Fa.2 and determination of their sequences and antioxidant activities. 

### 2.5. Amino Acid Compositions of Fa and Fa.2

The amino acid compositions of the fractions Fa and Fa.2 were investigated for evaluating the effect of composition on the antioxidant activity. As shown in Table 1, Glu, Gly, Ala, and Pro were the major amino acids in both Fa and Fa.2, which supported that collagen from sea cucumber was rich in Glu, Gly and Pro [15]. However, small amounts of Cys, Met, Ile, and His residues were identified in both Fa and Fa.2. After isolation of Fa by gel filtration, a significant increase in Glu, Ala, Tyr, and Phe was observed in the subfraction Fa.2. However, the contents of hydroxyl-containing amino acids (Thr and Ser) decreased in Fa.2 compared to Fa. The content of Gly decreased sharply from 295.5 in Fa to 238.5 residues/1000 residues in Fa.2. In addition, no significant differences were observed between the fractions Fa and Fa.2 in the contents of the sulfur-containing amino acid (Cys and Met) and some hydrophobic amino acid (Ile and Leu). Previous studies suggested that hydrophobic (Ala, Val, Ile, and Leu) and aromatic amino acids (Tyr and Phe) could promote the free radical scavenging activity of the peptides by direct electron transfer or interactions with hydrophobic targets, such as fatty acids and cell membrane [36,37,38]. The sulfur-containing amino acids (Met and Cys) and His were also regarded as potent amino acids contributing to the antioxidant property [15]. In addition, the antioxidant activities of peptides might also be improved if high contents of acidic amino acid residues (Asp and Glu) existed in the peptides [39]. Peptides with Pro possess intrinsic antioxidant activity [40]. In this study, the contents of Ile, Met, Cys and His were low in the fraction Fa.2. Therefore, the higher antioxidant activity of Fa.2 might be primarily attributed to the higher content of Ala, aromatic amino acids (Tyr and Phe), and Glu in the sequences.

### 2.6. Isolation of the Peptides from Fa.2 by RP-HPLC

Using ultrafiltration (UF) membrane and gel filtration chromatography, Fa.2 was isolated and subsequently purified by RP-HPLC. Four peptides (Fa.2-A, Fa.2-B, Fa.2-C, and Fa.2-D) were isolated (Figure 8a), and their antioxidant activities were determined. Figure 8b shows that Fa.2-C possessed the highest DPPH scavenging activity (35%) at the concentration of 0.2 mg/mL among the four subfractions, followed by Fa.2-A and Fa.2-D, whereas Fa.2-B showed the lowest DPPH scavenging activity (20%). Therefore, Fa.2-C was selected for determining the sequence by using a protein/peptide sequencer. Consequently, the sequence of Fa.2-C was determined to be Phe-Leu-Ala-Pro (FLAP), which scavenged DPPH radicals with the EC_50_ values of 0.385 mg/mL (data not shown). The EC_50_ of Fa.2-C was lower than that of HDHPVC (41.62 mg/mL) [41], YPPAK (2.62 mg/mL) [42], LMGQW (1.399 mg/mL) [43], PYFNK (4.11 mg/mL) [44], and YLMSR (2.74 mg/mL) [45] from protein hydrolysates of round scad, blue mussel, monkfish, scalloped hammerhead and croceine croaker, but higher than those of GPGGFI (0.194 mg/mL) and FIGP (0.118 mg/mL) from protein hydrolysate of bluefin leatherjacket skin [31]. A high DPPH radical scavenging activity of Fa.2-C indicated that it might reduce the oxidative damage in the body by converting DPPH radical to less harmful substances. In addition, the scavenging activities of other radicals, including hydroxyl and superoxide radicals, will be further studied.

## 3. Materials and Methods 

### 3.1. Chemicals and Reagents

Sea cucumber *Acaudina molpadioides* was provided from Zhoushan Jingzhou Aquatic Food Co., Ltd., in Zhejiang Province of China. The 2,2-dipehnyl-1-picryldydrazyl (DPPH), 2’-amino-bis-3-ethyl- benzothiazoline-6-sulphonic acid (ABTS), and protease (100,000 U/g) were purchased from Sigma-Aldrich (Shanghai, China). The ultrafiltration membrane (5 kDa and 1 kDa) was procured from Pall Corporation (New York, NY, USA). All other chemicals used were of analytical grade.

### 3.2. Extraction of Collagen from the body wall of Acaudina molpadioides

The body wall of *Acaudina molpadioides* was soaked in a 10-fold volume of 0.2 M EDTA (pH 8.0) for 48 h to remove the heavy metals. Subsequently, the EDTA solution was removed, and a 10-fold volume of 0.1 M NaOH was added for removing the non-collagenous proteins. After 48 h, the sample was centrifuged at 12,000 rpm for 30 min and washed with the deionized water to achieve the neutral pH. Then, Then, a 10-fold volume of 15% (*v*/*v*) butyl alcohol was added to the sediment and soaked for 48 h. The sediment was washed with the deionized water three times and then soaked in a 10-fold volume of 0.5 M acetic acid for 48 h. NaCl was added into the filtrate obtained by two layers of cotton cloth with the final concentration of 2.5 M. After centrifugation at 12,000 rpm for 30 min, the precipitate was collected and dissolved in a 0.5 M acetic acid solution. The solution was dialyzed against the deionized water for 24 h with a change of solution every 12 h. The collagen (ASC-Am) was obtained after lyophilizing the resulting dialysate. 

### 3.3. Preparation of Hydrolysates of Collagen ASC-Am

About 0.2 g of ASC-Am was added into 20 mL different buffer solutions and hydrolyzed for 30 min using each of the four proteases under the microwave irradiation (0, 50, 100, 150, 200, 250 and 300 W) in the XH-300A microwave-ultrasonic combined synthesis/extraction instrument (Beijing XiangHu Science and Technology Development Co., Ltd., Beijing, China). The enzyme/substrate ratio was 5% (w/w) and the reaction was carried out at the optimal temperatures and pH conditions for each protease as follows: papain (0.2 M phosphate, pH 6.5, 37 °C), pepsin (0.2 M Gly-HCl, pH 2.0, 37 °C), trypsin (0.2 M Tris-HCl, pH 8.5, 45 °C) and neutrase (0.2 M phosphate, pH 7.0, 55 °C ). Enzymes were inactivated in boiling water for 10 min. After centrifuging the hydrolysates at 12000 rpm for 30 min, the supernatants were fractionated based on molecular mass by two ultrafiltration membranes, cut-off 5 kDa and 1 kDa. Three fractions (Fa, Mw < 1 kDa; Fb, 1 kDa < Mw < 5 kDa; and Fc, Mw > 5 kDa) were collected separately, concentrated, freeze dried, and assayed for DPPH and ABTS radicals scavenging activity.

### 3.4. Determination of the Content of the Fractions

The contents of Fa, Fb, and Fc were estimated using HPLC (Agilent 1260 HPLC, Agilent Ltd., Palo Alto, USA) with a TSKgel 2000 SWXL at 220 nm. The mobile phase consisted of acetonitrile/water/trifluoroacetic acid (45:55:0.1, *v*/*v*/*v*) was run at a flow rate of 0.5 mL/min. Peroxidase (Mw 40,000 Da), aprotinin (Mw 6512 Da), clam peptide (Mw 1751.78 Da), and oligopeptide of anemone (Mw 531.61 Da) were used as the Mw markers.

### 3.5. Measurement of Antioxidant Activity

#### 3.5.1. DPPH Radical Scavenging Activity

Next, 2 mL of DPPH solution (0.04 mol/mL), 1 mL ethanol, and 1 mL of sample solution (1 mg/mL) were mixed together. The mixture was incubated for 30 min at room temperature and then centrifuged at 5000 rpm for 5 min. The control (2 mL DPPH solution + 1 mL ethanol + 1 mL water) and the blank (2 mL water + 1 mL ethanol + 1 mL sample solution) were also prepared. The absorbance of the sample, control and blank was measured at 517 nm. The antioxidant activity of the sample was evaluated by determining the scavenging rate of DPPH with the equation:DPPH scavenging rate (%) = (A_0_ − A + A_1_)/A_0_ × 100%
where A was the sample absorbance; A_0_ was the control absorbance; A_1_ was the blank absorbance.

#### 3.5.2. ABTS Radical Scavenging Activity

The stock ABTS radical solution was obtained by mixing 440 μL of potassium persulfate solution (140 mM) and 25 mL of ABTS solution (7 mM) and incubated in the dark for 14 hours. The stock solution was diluted with phosphate buffer (20 mM, pH 7.5) to an absorbance of 0.7 ± 0.002 as a working solution. The peptide sample solutions (1 mg/mL) were added to the working solution at a ratio of 1:4 (*v/v*). The mixture was incubated at the room temperature for 10 min, and then the absorbance was measured at 734 nm. The control without sample was also prepared and measured. The antioxidant activity of the sample was evaluated by determining the scavenging rate of ABTS with the following equation:ABTS scavenging rate (%) = (A_c_ − A_s_)/A_c_ × 100%
where A_c_ and A_s_ are the absorbances of the control and sample, respectively. 

### 3.6. Gel Filtration Chromatography of Fa

Fa solution (2 mL, 60.0 mg/mL) was separated on a Sephadex G-15 (Sigma-Aldrich, Shanghai, China) column (1 × 100 cm). We used a 0.02 M phosphate (pH 7.0) buffer at the flow rate of 0.15 mL/min for the elution. The eluate was monitored at 280 nm and each fraction was collected and lyophilized. 

### 3.7. Amino Acid Analysis

The samples of fractions Fa and Fa.2 were hydrolyzed with 6 N HCl at 110 °C for 24 h. The amino acid composition analyses of the hydrolysates were performed using an automated amino acid analyzer of HITACHI L8900 (Hitachi High-Technologies Corporation, Tokyo, Japan).

### 3.8. RP-HPLC and Identification of Peptide Sequences

Fa.2 was dissolved in TFA (0.1%, *v*/*v*) with the final concentration of 0.1 mg/mL. 20 µL Fa.2 solution was injected into a Zorbax SB-C18 column, and further separated and determined at 280 nm by RP-HPLC (Agilent 1260 HPLC, Agilent Ltd., Palo Alto, USA) with a linear gradient of ACN (0–75% *v*/*v*, 45 min) at a flow rate of 0.8 mL/min. Four peptides (defined as Fa.2-A, Fa.2-B, Fa.2-C, and Fa.2-D) were collected, and their DPPH radical scavenging activities were measured. Fa.2-C was subjected to N-terminal amino acid sequencing on an Applied Biosystems 494 protein sequencer (Thermo Fisher Scientific (China) Co., Ltd., Shanghai, China).

### 3.9. Statistical Analysis

All experiments were performed in triplicate. Significant differences (*p* < 0.05) between means were identified by Duncan’s multiple range test. Results were analyzed using the software SPSS 19.0 (SPSS Inc., Chicago, IL, USA).

## 4. Conclusions

The present research showed that microwave-assisted enzymatic hydrolysis might be a promising method for preparing the antioxidant peptides from the collagen of *Acaudina molpadioides* (ASC-Am) obtained from Zhejiang Province in China. Neutrase was the best enzyme based on the highest content and antioxidant activity of the peptide with a low molecular weight (Mw < 1 kDa). After 30 min of reaction and at 300 W of the microwave power, the content of the fraction Fa (Mw < 1 kDa) exceeded 75%, and the ABTS scavenging activity of Fa was over 80% at the concentration of 2 mg/mL. The fraction Fa.2 was obtained from the fraction Fa by gel filtration, which showed high DPPH scavenging activity. The amino acid composition analysis indicated that Fa.2 possessed higher contents of aromatic (Tyr and Phe) and acidic amino acids (Glu). Fa.2 was further isolated by RP-HPLC, and one of the fractions was identified as Phe-Leu-Ala-Pro with EC_50_ value of 0.385 mg/mL. The findings of this research may inspire the application of sea cucumber *Acaudina molpadioides* in biomedicines and functional foods.

## Figures and Tables

**Figure 1 marinedrugs-17-00169-f001:**
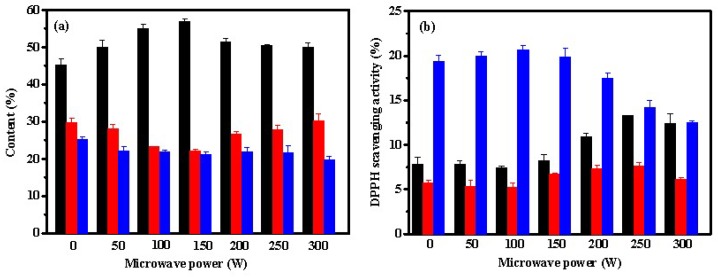
Effect of microwave power on contents (**a**) and DPPH scavenging activities (**b**) of Fa (■), Fb (■), and Fc (■) from hydrolysate of ASC-Am by papain. The experiment was performed in triplicate within three days with the same collagen obtained from the same batch of sea cucumber. All values were the mean ± standard deviation (SD).

**Figure 2 marinedrugs-17-00169-f002:**
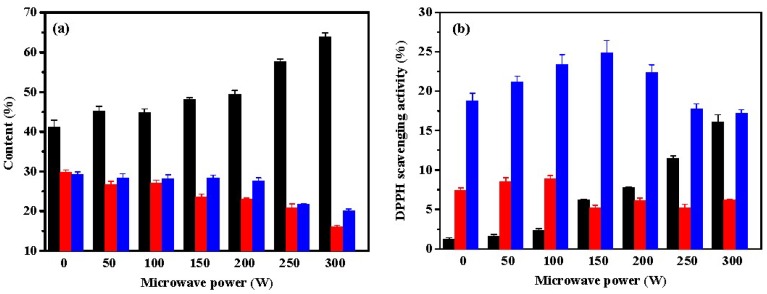
Effect of microwave power on contents (**a**) and DPPH scavenging activities (**b**) of Fa (■), Fb (■), and Fc (■) from hydrolysate of ASC-Am by pepsin. The experiment was performed in triplicate within three days with the same collagen obtained from the same batch of sea cucumber. All values were the mean ± standard deviation (SD).

**Figure 3 marinedrugs-17-00169-f003:**
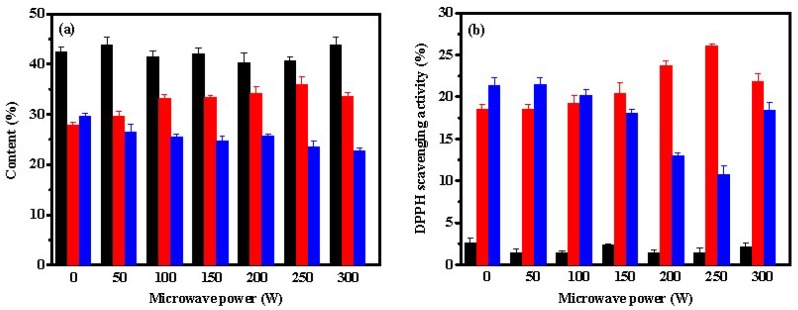
Effect of microwave power on contents (**a**) and DPPH scavenging activities (**b**) of Fa (■), Fb (■), and Fc (■) from hydrolysate of ASC-Am by trypsin. The experiment was performed in triplicate within three days with the same collagen obtained from the same batch of sea cucumber. All values were the mean ± standard deviation (SD).

**Figure 4 marinedrugs-17-00169-f004:**
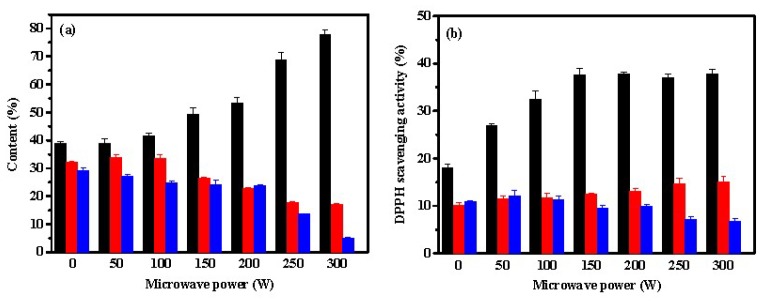
Effect of microwave power on contents (**a**) and DPPH scavenging activities (**b**) of Fa (■), Fb (■), and Fc (■) from hydrolysates of ASC-Am by neutrase. The experiment was performed in triplicate within three days with the same collagen obtained from the same batch of sea cucumber. All values were the mean ± standard deviation (SD).

**Figure 5 marinedrugs-17-00169-f005:**
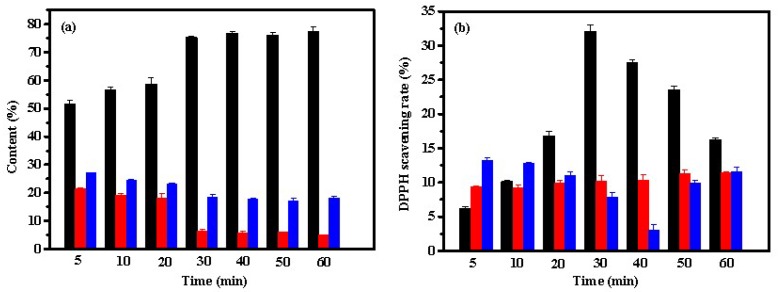
Effect of microwave time on contents (**a**) and DPPH scavenging activities (**b**) of Fa (■), Fb (■), and Fc (■) from hydrolysates of ASC-Am by neutrase. The experiment was performed in triplicate within three days with the same collagen obtained from the same batch of sea cucumber. All values were the mean ± standard deviation (SD).

**Figure 6 marinedrugs-17-00169-f006:**
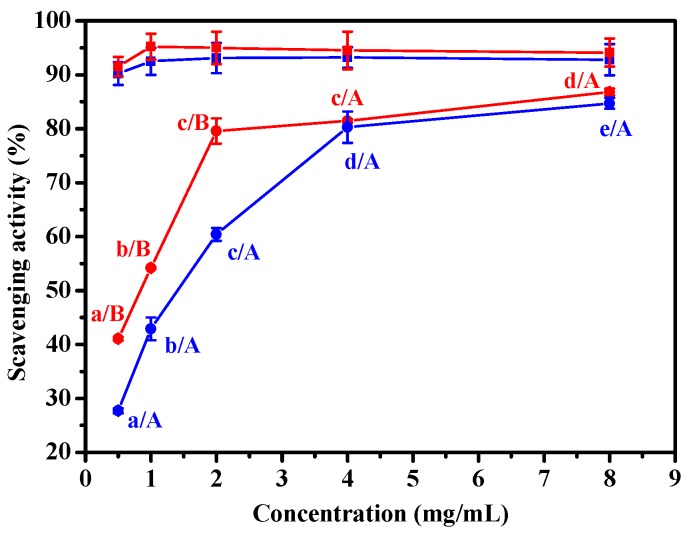
Effect of Fa (●, ●) and ascorbic acid (■, ■) concentrations on DPPH radical (●, ■) and ABTS radical (●, ■) scavenging activity. Fa was the hydrolysate (Mw < 1 kDa) by neutrase hydrolysis of ASC-Am at 300 W for 30 min. The experiment was performed in triplicate within three days with the same sample of Fa. All values were the mean ± standard deviation (SD). (a–e) Values with different letters indicated significant differences in the same free radical at different concentrations (p < 0.05); (A–B) Values with different letters indicated significant differences in the different free radical at the same concentration (p < 0.05).

**Figure 7 marinedrugs-17-00169-f007:**
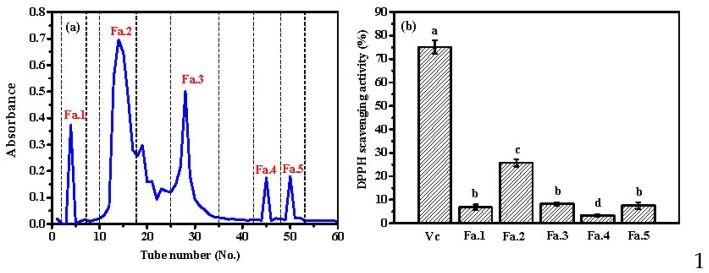
(**a**) Separation scheme of antioxidant peptide from Fa by Sephadex G-25 column; (**b**) The DPPH scavenging activity of fractions from Fa at concentration of 0.2 mg/mL. Fa was the hydrolysate (Mw < 1 kDa) by neutrase hydrolysis of ASC-Am at 300 W for 30 min. The experiment was performed in triplicate within three days with the same sample of Fa. All values were the mean ± standard deviation (SD). (a–d) Values with different letters indicated significant differences in the different sample at same concentrations (*p <* 0.05).

**Figure 8 marinedrugs-17-00169-f008:**
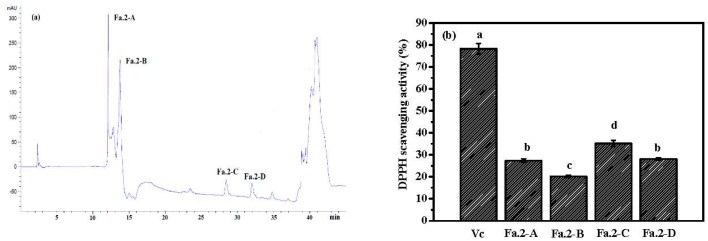
RP-HPLC chromatography of Fa.2 on Zorbax. SB C-18 column (**a**) and the DPPH scavenging activity of four peptides at concentration of 0.2 mg/mL (b). Fa.2 was isolated from the hydrolysate Fa obtained by neutrase hydrolysis of ASC-Am at 300 W for 30 min. The experiment was performed in triplicate within three days with the same sample of Fa.2. (a–d) Values with different letters indicated significant differences in the different sample at same concentrations (*p <* 0.05).

**Table 1 marinedrugs-17-00169-t001:** Amino acid composition of Fa and Fa.2 (residues/1000). Fa was the hydrolysate (Mw < 1 kDa) by neutrase hydrolysis of ASC-Am at 300 W for 30 min. The experiment was performed in triplicate within three days with the same sample of Fa and Fa.2. All values were the mean ± standard deviation (SD).

Amino Acid	Fa	Fa.2
Aspartic acid (Asp)	64.3 ± 2.5	68.3 ± 2.4
Threonine (Thr)	39.5 ± 0.8	30.6 ± 1.6
Serine (Ser)	48.3 ± 1.4	35.2 ± 1.1
Glutamic acid (Glu)	102 ± 3.2	138.5 ± 4.2
Glycine (Gly)	295.5 ± 4.2	238.5 ± 4.8
Alanine (Ala)	125.1 ± 2.5	154.2 ± 3.1
Cysteine (Cys)	7.8 ± 0.4	7.3 ± 0.2
Valine (Val)	15.5 ± 0.8	15.5 ± 0.8
Methionine (Met)	7.8 ± 0.5	6.4 ± 0.2
Isoleucine (Ile)	5.6 ± 0.3	5.4 ± 0.2
Leucine (Leu)	19.2 ± 0.5	18.6 ± 1.1
Tyrosine (Tyr)	9.6 ± 0.6	18.5 ± 0.5
Phenylalanine (Phe)	10.3 ± 0.5	18.5 ± 0.9
Lysine (Lys)	10.4 ± 0.6	11.1 ± 0.5
Histidine (His)	5.6 ± 0.3	5.9 ± 0.2
Arginine (Arg)	58.6 ± 1.8	52.6 ± 1.8
Proline (Pro)	106.6 ± 3.5	108.3 ± 2.4
Hydroxyproline (Hyp)	68.2 ± 2.0	65.8 ± 2.0

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
