# Peer review of "Preparation and Evaluation of Peptides with Potential Antioxidant Activity by Microwave Assisted Enzymatic Hydrolysis of Collagen from Sea Cucumber Acaudina Molpadioides Obtained from Zhejiang Province in China"

_marinedrugs, 2019, doi:10.3390/md17030169_

Round 1

Reviewer 1 Report

The manuscript is revised according to the comments and can be accepted.

Author Response

Thanks

Reviewer 2 Report

The manuscript "Preparation and Properties of Antioxidative Peptides with Microwave Assisted Enzyme Hydrolysis of Collagen from Sea Cucumber Acaudina molpadioides" has improved since the previous submission but there still several concerns and flaws to fix. It seems that improvement has been focused in English as the experimental data are the same. When an article is rejected and asked to resubmit with new experiments, more changes are expected than improving English and adding one experiment when reproducibility concerns have been given. And a letter / comment of what has been changed from prior submission should be provided.

Major concerns:

-It is stated in line 418 that all experiments were performed in triplicate. However, it would be more desirable to specify in each experiment how triplicates were done. For example, different days, different samples of the sea cucumber (from same or different vendor/localization), and so on.

-This concern from previous submission about reproducibility  "There are serious doubts regarding the reproducibility of the results reported by the authors. If I have understood well (and if Material and Methods are correct) the Figure 5B shows the antioxidant properties of the Fraction a (also b and c) after enzymatic cleavage with Neutrase assisted by microwave radiation at 300 W. After 1 h of MW treatment, the DPPH scavenging activity is between 15 and 20%. The same conditions apply for Figure 4B... and there the DPPH scavenging rate for the same concentration, intensity and time is around 40%. To enworse the reproducibility issue, Figure 6 b should be the same than Figure 5B (same conditions) but they are completely different. These discrepancies are very serious flaws and arises logical doubts regarding the credibility and reproducibility of the reported experiments. " have not been addressed. Please address.

-In section 2.3 it is stated that analysis is done for digestion with neutrase. In following sections (2.4, 2.5, 2.6) this should be remembered, and in addition, the microwave power should be specified in all cases.

-The enzymatic digestion reproducibility needs to be proved, for example, providing data of repetitions in Supplementary material. It sounds rather strange that such a complex process as an enzymatic digestion has such low standard deviation. Fragment patterns should be compared to see the reproducibility of the results. HPLC has been done for Fa, but only for the second repetition. First and third could be provided in Supplementary, as well as Fb and Fc. This is in line of the non-addressed concern of reproducibility. Additional experiments as a Western blot can be also added.

-Positive control for DPPH assay (for instance, Vitamin C) has been added but only in one analysis. This positive control could be added, for comparison issues, to all the remaining figures.

-Authors should do the same experiment with a marine sample of Acaudina olpadioides from a different source (vendor or localization) to verify that this pattern of fragmentation is observed in organisms of independent locations or if it is specific of samples picked in a certain place.

-Why formats of Figure 5 is different of those of figures 1-4. Please homogenise.

-It is logical that the aminoacid composition of Fa and Fa2 are similar as Fa2 is the biggest fraction according to HPLC results. The composition of other fractions could be determined to compare: here differences should be seen and this could give hints about what peptides are responsible of the DPPH activity (but we should keep in mind that a 25% of DPPH radical scavenging activity is low).

-This concern from previous submission has not been addressed. "3) Relevance of the antioxidant activity results Regarding the significance of the antioxidant results, the antioxidant activity of the majority of the fractions is below 20%, with can be considered as low, as none of these compounds has at least a 50% of the activity of the control, which starts to be considered a good antioxidant activity."

Minor concerns:

-It would help the readers of Nature Drugs with less experience in Zoology to have images of the sea cucumbers used. It would also increase the quality of the paper.

English:

English has significantly improved. However, a few typos can be detected; for example "medicical" (line 46), and in the title “antioxidant” would be better than “antioxidative”. Please revise all manuscript.

Author Response

Title: Preparation and Evaluation of Antioxidant Peptides with Microwave Assisted Enzymatic Hydrolysis of Collagen from Sea Cucumber Acaudina molpadioides

Manuscript ID: marinedrugs-456621

Dear reviewer,

Thanks again for your valuable comments. We have revised the manuscript according to the comments that we received. The modified part of the text has been marked in red. In addition, the point-by-point response to the reviewer’s comments were provided. We hope the revised manuscript and response will meet your standard, thanks.

As to comments of Reviewer:

(1) The manuscript "Preparation and Properties of Antioxidative Peptides with Microwave Assisted Enzyme Hydrolysis of Collagen from Sea Cucumber Acaudina molpadioides" has improved since the previous submission but there still several concerns and flaws to fix. It seems that improvement has been focused in English as the experimental data are the same. When an article is rejected and asked to resubmit with new experiments, more changes are expected than improving English and adding one experiment when reproducibility concerns have been given. And a letter /comment of what has been changed from prior submission should be provided.

    Answer: Compared to the original manuscript, the revised manuscript we resubmitted last time has been greatly modified, for example, we removed the infrared spectrum of Fa and Fig 6b. In addition, we also added some experimental data, such as the DPPH and ABTS scavenging activity of ascorbic acid, the separation of the Fa.2 by the reverse phase liquid chromatography. All modifications were also marked in red in the revised manuscript we resubmitted last time. A comment with a point-by-point response to the reviewer’s comments was also provided. Of course, we are very sorry that some experiments had not been added in time for some special reasons. Thank you for understanding.

(2) It is stated in line 418 that all experiments were performed in triplicate. However, it would be more desirable to specify in each experiment how triplicates were done. For example, different days, different samples of the sea cucumber (from same or different vendor/localization), and so on.

Answer: The repeated experiments were performed by the same sample within three days. We have added the description in title of each figure and table, thanks.

(3) This concern from previous submission about reproducibility  "There are serious doubts regarding the reproducibility of the results reported by the authors. If I have understood well (and if Material and Methods are correct) the Figure 5B shows the antioxidant properties of the Fraction a (also b and c) after enzymatic cleavage with neutrase assisted by microwave radiation at 300 W. After 1 h of MW treatment, the DPPH scavenging activity is between 15 and 20%. The same conditions apply for Figure 4B... and there the DPPH scavenging rate for the same concentration, intensity and time is around 40%. To enworse the reproducibility issue, Figure 6 b should be the same than Figure 5B (same conditions) but they are completely different. These discrepancies are very serious flaws and arises logical doubts regarding the credibility and reproducibility of the reported experiments. " have not been addressed. Please address.

Answer: The experiment of Figure 4b was performed with neutrase assisted by microwave radiation at 300 W for 30 min, not 1 h (section 3.3). We had explained this problem in paper “answer the comments of reviewer” which we resubmitted last time. In addition, the conditions of Figure 5b and Figure 6b were different. The time in Figure 5b was the time of MW treatment, but the time in Figure 6b was the time of scavenging DPPH or ABTS radicals at the antioxidant assay (section 3.5.1 and 3.5.2), not the MW treatment. After referring to many researches, we think that Figure 6b is not necessary, so it had been deleted in the revised manuscript we resubmitted last time.

(4)In section 2.3 it is stated that analysis is done for digestion with neutrase. In following sections (2.4, 2.5, 2.6) this should be remembered, and in addition, the microwave power should be specified in all cases.

Answer: Thanks for your suggestion. The digestion with neutrase was added and remembered in sections 2.4, 2.5, and 2.6. The microwave power was also specified in sections 2.4, 2.5, and 2.6.

(5) The enzymatic digestion reproducibility needs to be proved, for example, providing data of repetitions in Supplementary material. It sounds rather strange that such a complex process as an enzymatic digestion has such low standard deviation. Fragment patterns should be compared to see the reproducibility of the results. HPLC has been done for Fa, but only for the second repetition. First and third could be provided in Supplementary, as well as Fb and Fc. This is in line of the non-addressed concern of reproducibility. Additional experiments as a Western blot can be also added.

Answer: Each experiment was repeated with the same sample within three days. For example, the sample of three experiments in Figure 1 (also in Figure 2-4) was the same collagen which was prepared from the same batch of sea cucumber. I don't understand what means of the sentence “HPLC has been done for Fa, but only for the second repetition. First and third could be provided in Supplementary, as well as Fb and Fc”. The amino acid compositions of Fa by HPLC were performed three time. The average value and standard deviation are listed in Table 1. The Separation of Fa.2 by HPLC(Figure 8) was also done three times. So I don't understand what means of “but only for the second repetition”. The experiment of Western blot is being done, but we are sorry that the result would not be provided due to the deadline of 5 days.

(6) Positive control for DPPH assay (for instance, Vitamin C) has been added but only in one analysis. This positive control could be added, for comparison issues, to all the remaining figures.

Answer: According to the comments of reviewer, we added the positive control of Vitamin C in Figure 7b and 8b, thanks.

(7) Authors should do the same experiment with a marine sample of Acaudina olpadioides from a different source (vendor or localization) to verify that this pattern of fragmentation is observed in organisms of independent locations or if it is specific of samples picked in a certain place.

Answer: In order to reduce the experimental error, we used the same sample of Acaudina molpadioides from the same company for each experiment. The Acaudina molpadioides is found in various areas of the East China Sea. The amino acid composition of collagen of Acaudina molpadioides in different sea areas will be different. Therefore, we do not think it is necessary to do the same experiment using the Acaudina molpadioides from different regions. Thanks for the suggestion of reviewer.

(8) Why formats of Figure 5 is different of those of figures 1-4. Please homogenise.

Answer: According to the comments of reviewer, Figure 5 was changed to the bar chart, thanks.

(9) It is logical that the amino acid composition of Fa and Fa2 are similar as Fa2 is the biggest fraction according to HPLC results. The composition of other fractions could be determined to compare: here differences should be seen and this could give hints about what peptides are responsible of the DPPH activity (but we should keep in mind that a 25% of DPPH radical scavenging activity is low).

Answer: Thanks for the comments of reviewer. The composition of other fractions could be determined in further study. However, we are very sorry that the result would not be provided in time due to the deadline of 5 days.

(10) This concern from previous submission has not been addressed. "3) Relevance of the antioxidant activity results Regarding the significance of the antioxidant results, the antioxidant activity of the majority of the fractions is below 20%, with can be considered as low, as none of these compounds has at least a 50% of the activity of the control, which starts to be considered a good antioxidant activity."

Answer: Thanks for the reviewer’s comments. The antioxidant activity of the majority of the fractions is below 20%, mainly because the sample concentration is relatively low. For example, the antioxidant activity of Fa.1-5 was measured at concentration of 0.2 mg/mL. The antioxidant activity of Fa.2 was over 80% at concentration of 1 mg/mL.

(11) It would help the readers of Nature Drugs with less experience in Zoology to have images of the sea cucumbers used. It would also increase the quality of the paper.

Answer: Thanks for the comments of reviewer. We will provide the images of the sea cucumbers used in Supplementary, thanks.

(11) English has significantly improved. However, a few typos can be detected; for example "medicical" (line 46), and in the title “antioxidant” would be better than “antioxidative”. Please revise all manuscript.

Answer: Thanks for the comments of reviewer. We have corrected the mistakes in revised manuscript.

Reviewer 3 Report

The authors have replied to all my questions and doubts, and added my suggestions. I believe the manuscript can be published in Marine Drugs.

Author Response

Tanks.

Round 2

Reviewer 2 Report

Dear Authors,

Please find my responses to your answers in blue colour.

As general comment, I cannot accept manuscript if it states that doing the described diggestion antioxidant peptides from Acaudina molpadioides are obtained. Authors also state (in discussion / abstract / conclusions) that if they do the diggestion with a different sample, from other region / provider, the composition of the collagen will vary and then has no point in repeating as the results are different: this is a recognition of the lack of reproducibility. This fact suggest that changing that may result in a change also in the antioxidant activity.

Seeing that manuscript has been accepted by the remaining reviewers, we can do an intermediate solution to fix this big flaw.

To help and to favour the publication I propose to modify the title, for example to

"Preparation and Evaluation of Peptides with potential antioxidant activity by Microwave Assisted Enzymatic Hydrolysis of Collagen from Sea Cucumber Acaudina molpadioides obtained from Zhejiang Province in China"

This title would be more specific and would not imply a total generalization, which would be incorrect due to the lack of variation in the samples. In this case it is stated that results are restricted to this specific samples, and then I would accept the manuscript: the lack of generalization would imply that results apply to a specific sample, and that authors want to report these specific findings. This is acceptable for publication. The current focus, not.

In line to this title change, I encourage Authors to revise abstract, discussion and conclusions to introduce this specific point of view.

(1) The manuscript "Preparation and Properties of Antioxidative Peptides with Microwave Assisted Enzyme Hydrolysis of Collagen from Sea Cucumber Acaudina molpadioides" has improved since the previous submission but there still several concerns and flaws to fix. It seems that improvement has been focused in English as the experimental data are the same. When an article is rejected and asked to resubmit with new experiments, more changes are expected than improving English and adding one experiment when reproducibility concerns have been given. And a letter /comment of what has been changed from prior submission should be provided.

Answer: Compared to the original manuscript, the revised manuscript we resubmitted last time has been greatly modified, for example, we removed the infrared spectrum of Fa and Fig 6b. In addition, we also added some experimental data, such as the DPPH and ABTS scavenging activity of ascorbic acid, the separation of the Fa.2 by the reverse phase liquid chromatography. All modifications were also marked in red in the revised manuscript we resubmitted last time. A comment with a point-by-point response to the reviewer’s comments was also provided. Of course, we are very sorry that some experiments had not been added in time for some special reasons. Thank you for understanding.

The general comment given above answers this.

(2) It is stated in line 418 that all experiments were performed in triplicate. However, it would be more desirable to specify in each experiment how triplicates were done. For example, different days, different samples of the sea cucumber (from same or different vendor/localization), and so on.

Answer: The repeated experiments were performed by the same sample within three days. We have added the description in title of each figure and table, thanks.
Thanks for your answer, concern fixed.

(3) This concern from previous submission about reproducibility "There are serious doubts regarding the reproducibility of the results reported by the authors. If I have understood well (and if Material and Methods are correct) the Figure 5B shows the antioxidant properties of the Fraction a (also b and c) after enzymatic cleavage with neutrase assisted by microwave radiation at 300 W. After 1 h of MW treatment, the DPPH scavenging activity is between 15 and 20%. The same conditions apply for Figure 4B... and there the DPPH scavenging rate for the same concentration, intensity and time is around 40%. To enworse the reproducibility issue, Figure 6 b should be the same than Figure 5B (same conditions) but they are completely different. These discrepancies are very serious flaws and arises logical doubts regarding the credibility and reproducibility of the reported experiments. " have not been addressed. Please address.

Answer: The experiment of Figure 4b was performed with neutrase assisted by microwave radiation at 300 W for 30 min, not 1 h (section 3.3). We had explained this problem in paper “answer the comments of reviewer” which we resubmitted last time. In addition, the conditions of Figure 5b and Figure 6b were different. The time in Figure 5b was the time of MW treatment, but the time in Figure 6b was the time of scavenging DPPH or ABTS radicals at the antioxidant assay (section 3.5.1 and 3.5.2), not the MW treatment. After referring to many researches, we think that Figure 6b is not necessary, so it had been deleted in the revised manuscript we resubmitted last time.
Thanks for your answer, understood.

(4)In section 2.3 it is stated that analysis is done for digestion with neutrase. In following sections (2.4, 2.5, 2.6) this should be remembered, and in addition, the microwave power should be specified in all cases.

Answer: Thanks for your suggestion. The digestion with neutrase was added and remembered in sections 2.4, 2.5, and 2.6. The microwave power was also specified in sections 2.4, 2.5, and 2.6.
Thanks for your answer, concern fixed.

(5) The enzymatic digestion reproducibility needs to be proved, for example, providing data of repetitions in Supplementary material. It sounds rather strange that such a complex process as an enzymatic digestion has such low standard deviation. Fragment patterns should be compared to see the reproducibility of the results. HPLC has been done for Fa, but only for the second repetition. First and third could be provided in Supplementary, as well as Fb and Fc. This is in line of the non-addressed concern of reproducibility. Additional experiments as a Western blot can be also added.

Answer: Each experiment was repeated with the same sample within three days. For example, the sample of three experiments in Figure 1 (also in Figure 2-4) was the same collagen which was prepared from the same batch of sea cucumber. I don't understand what means of the sentence “HPLC has been done for Fa, but only for the second repetition. First and third could be provided in Supplementary, as well as Fb and Fc”. The amino acid compositions of Fa by HPLC were performed three time. The average value and standard deviation are listed in Table 1. The Separation of Fa.2 by HPLC(Figure 8) was also done three times. So I don't understand what means of “but only for the second repetition”. The experiment of Western blot is being done, but we are sorry that the result would not be provided due to the deadline of 5 days.
Thanks for your answer. The given concern is in line of the change - in title - asked to approve article. It would have been desirable to repeat with different samples, but I understand that authors want to report these specific findings. So, in this case, title, abstract and conclusions should be ammended in accordance to this specificity. If western blot results are obtained in these time it should be added to the manuscript.

(6) Positive control for DPPH assay (for instance, Vitamin C) has been added but only in one analysis. This positive control could be added, for comparison issues, to all the remaining figures.

Answer: According to the comments of reviewer, we added the positive control of Vitamin C in Figure 7b and 8b, thanks.
Thanks for your answer, concern fixed. It would have been desirable for all figures but I understand that for the remaining it would have implied a more complicated legend in axis. 

(7) Authors should do the same experiment with a marine sample of Acaudina olpadioides from a different source (vendor or localization) to verify that this pattern of fragmentation is observed in organisms of independent locations or if it is specific of samples picked in a certain place.

Answer: In order to reduce the experimental error, we used the same sample of Acaudina molpadioides from the same company for each experiment. The Acaudina molpadioides is found in various areas of the East China Sea. The amino acid composition of collagen of Acaudina molpadioides in different sea areas will be different. Therefore, we do not think it is necessary to do the same experiment using the Acaudina molpadioides from different regions. Thanks for the suggestion of reviewer.
In this case it would have been better to repeat the diggestion with different samples of different providers / regions. Doing solely with one sample does not warrant that all the times this diggestion is applied, antioxidant peptides will be obtained, because as Authors state, their composition will be different.

So an intermediate solution could be modifying the title, abstract and conclusions as suggested in the general comment.

(8) Why formats of Figure 5 is different of those of figures 1-4. Please homogenise.

Answer: According to the comments of reviewer, Figure 5 was changed to the bar chart, thanks.
Thanks for your answer, concern fixed.

(9) It is logical that the amino acid composition of Fa and Fa2 are similar as Fa2 is the biggest fraction according to HPLC results. The composition of other fractions could be determined to compare: here differences should be seen and this could give hints about what peptides are responsible of the DPPH activity (but we should keep in mind that a 25% of DPPH radical scavenging activity is low).

Answer: Thanks for the comments of reviewer. The composition of other fractions could be determined in further study. However, we are very sorry that the result would not be provided in time due to the deadline of 5 days.
Thanks for your answer, this concern would be fixed if the asked changes are introduced

(10) This concern from previous submission has not been addressed. "3) Relevance of the antioxidant activity results Regarding the significance of the antioxidant results, the antioxidant activity of the majority of the fractions is below 20%, with can be considered as low, as none of these compounds has at least a 50% of the activity of the control, which starts to be considered a good antioxidant activity."

Answer: Thanks for the reviewer’s comments. The antioxidant activity of the majority of the fractions is below 20%, mainly because the sample concentration is relatively low. For example, the antioxidant activity of Fa.1-5 was measured at concentration of 0.2 mg/mL. The antioxidant activity of Fa.2 was over 80% at concentration of 1 mg/mL.
Thanks for your answer. I would not say that a concentration of 0.2 mg/mL is low, this imply 200 mg/L. If if were, instead of a protein, a standard organic compound of MW of 500 Da (upper limit of Lipinski's Rule of Five), its concentration would be 400 μM, which is not low at all. The concentration of 1 mg/mL (the one that gives an activity closer to Vitamin C) would imply in this theoretical exercise a concentration of 2 mM, which is extremely high for common antioxidants.

Concluding, this supposed low activity should be mentioned accordingly in manuscript to provide a more real view of the reported data.

(11) It would help the readers of Nature Drugs with less experience in Zoology to have images of the sea cucumbers used. It would also increase the quality of the paper.

Answer: Thanks for the comments of reviewer. We will provide the images of the sea cucumbers used in Supplementary, thanks.
Thanks for taking into consideration this suggestion.

(11) English has significantly improved. However, a few typos can be detected; for example "medicical" (line 46), and in the title “antioxidant” would be better than “antioxidative”. Please revise all manuscript.

Answer: Thanks for the comments of reviewer. We have corrected the mistakes in revised manuscript.
Thanks for your answer, concern fixed.

Author Response

Thanks again for your comments. We have changed the title according to your suggestion. In addition, we also introduced the specific sample in abstract, discussion and conclusion. We are so sorry that the experiments of western blot and composition of other fractions had not been finished. The modified part of the text has been marked in red. We hope the revised manuscript will meet your standard. Thank you again for your tolerance and understanding.

Round 3

Reviewer 2 Report

Dear Authors,

Thank you for taking into account the suggestions given during reviewing process. Manuscript can be published in its present form.

Kind regards

This manuscript is a resubmission of an earlier submission. The following is a list of the peer review reports and author responses from that submission.

Round 1

Reviewer 1 Report

Dear Authors, 

This paper may be publishable in the Marine Drugs, since it has scientifical quality and presents some novelty, filling in fact a void in scientific knowledge concerning pharmaceutical and health food industries. Moreover, the presentation quality is also fine and the main ideas are clearly understandable. The paper is not long and main issues are approached and treated in a succinct manner.

However, the discussion of the results is frankly underdeveloped. In my point of view it’s essential to compare your results with other authors. Accordingly, I recommend its publication with important modifications, mainly, directed to development of the discussion. 

I have also some specific comments:

 -         Page 2, line 84: Did you determine the degree of hydrolysis of the different hydrolysates? I think this provide an important information about the extent of hydrolysis. You presented the content (%), what did it means and how did you calculate it?

 -         Page 2, line 87: “The effects of different microwave power on contents … by papain were studied” This sentence is repeated (lines 84-86).

 -         Figures 1 to 5: I think a bar chart its better in this case. 0W is the tradicional process?

 -         Page 6, Antioxidant activity of Fa: Please refer the EC50 value in the discussion.

 -         Page 6, line 207: Why did you test antioxidant activities at the concentration of 1.0 mg/ml?

 -         Page 7, line 257: Why did you test DPPH radical scavenging activity at the concentration of 0.2 mg/ml?

 -         Page 8, line 264, Amino acid composition of Fa and Fa.2: Did you determine the HYP content. In the case of collagen this amino acid is important.

 -         Page 9, line 303: DPPH and ABTS must be written out.

 -         Page 9, line 303: “protease” – which protease? You used papain, pepsin, trypsin and neutrase...

 -         Page 9, line 318: “…under different microwave power.” Please indicate the different microwave power.

 Best regards,

Carla Pires

Reviewer 2 Report

The manuscript is about the effect of microwave on the enzymatic hydrolysis of sea cucumber collagen and antioxidant activity of the hydrolysate. The method section of the manuscript is weakly written. Although the manuscript is about the effect of microwave on collagen hydrolysis any detail about the used microwave and the process of microwave aid hydrolysis and its setup is not mentioned. Since microwave can increase temperature and the temperature reduces enzyme activity it should be clarified that how this was controlled during the hydrolysis. The method section must be fully revised and clearly elaborated. The manuscript has no novelty according to the references mentioned in the introduction of the manuscript. Amino acid composition presented for the collagen hydrolysate does not show hydroxyproline which is one of the major amino acids in collagen which must be measured.  It is clear that protein hydrolysate contains higher amount of NH2 and COOH sine it hydrolyzed and it does not need confirmation with FTIR. However, the changes in the FTIR spectra of the hydrolysate to confirm higher content of NH2 and COOH does not seem logical. It is not clear that how content of each molecular weight fraction is calculated which should be also mentioned.

Reviewer 3 Report

The manuscript "Preparation and Properties of Antioxidative Peptides with Microwave Assisted Enzyme Hydrolysis of Collagen from Sea Cucumber Acaudina molpadioides" should not be published in its current form in Marine Drugs. The article has very serious flaws:

Major concerns: serious flaws

1) Reproducibility

There are serious doubts regarding the reproducibility of the results reported by the authors. If I have understood well (and if Material and Methods are correct) the Figure 5B shows the antioxidant properties of the Fraction a (also b and c) after enzymatic cleavage with Neutrase assisted by microwave radiation at 300 W. After 1 h of MW treatment, the DPPH scavenging activity is between 15 and 20%. The same conditions apply for Figure 4B... and there the DPPH scavenging rate for the same concentraton, intensity and time is around 40%. To enworse the reproducibility issue, Figure 6 b should be the same than Figure 5B (same conditions) but they are completely different. These discrepancies are very serious flaws and arises logical doubts regarding the credibility and reproducibility of the reported experiments.

To solve this issue:

-Authors should repeat the cleavage three times and evaluate if the content (Fractions a / b / c) are repeated in the three experiments.

-Authors should do the same experiment with a marine sample of Acaudina molpadioides from a diferent source (vendor or localization) to verify that this pattern of fragmentation is observed in organisms of independent locations or if it is specific of samples picked in a certain place.

2) Positive control and statistical significance

There is no positive control directly cited for the DPPH experiment. It should be compared with the activity of a known antioxidant as Vitamin C - ascorbic acid- or Vitamin E - tocopherol. In Material and Methods a control A0 (absorbance of this control) is cited, but it is not explained what compound is this control.

Besides, no error / standard deviation... is provided for the results.

3) Relevance of the antioxidant activity results

Regarding the significance of the antioxidant results, the antioxidant activity of the majority of the fractions is below 20%, with can be considered as low, as none of these compounds has at least a 50% of the activity of the control, which starts to be considered a good antioxidant activity.

Other concerns

1) English

English needs a deep revision and correction/rewriting by a native speaker as it is very poor. There are many sentences gramatically incorrect that need to be revised; there a non-concordances subject-verb; some singulars/plurals are wrongly used, the usage of articles is very deficient, and many typos can be found. One example: antioxidative should be replaced by antioxidant along all manuscript.

2) Minor concerns

Specific mistakes are seen in the introduction:

-Line 34: the reported [in bibliography] collagen peptide needs to be described or named.

-Lines 37-39: among the cited radicals, "peroxyl" is incorrect - should be corrected by hydrogen peroxide. Besides, ABTS and DPPH are not biological radicals: they are synthetic radicals that are used in antioxidant evaluation experiments as they have higher stability and thus can be stored in reproducible conditions. They should be cited in a differentiate manner taking this into account, and could be replaced by true ROS and RNS species as peroxynitrite radical and singlet oxygen.

And in the results and discussion:

-In Figure 4B a plateau in the antioxidant activity is observed at a microwave power >150 W. This has two direct consequences:

*First, it should be commented in the text, as it is relevant

*Second, this arises doubts about why this power has not been chosen: it is less aggresive for the collagen and provides the same antioxidant results, thus it could be more adequate for the subsequent experiments (microwave time, FTIR...)

-FTIR spectra of fractions B and C should be added for comparation issues.

Reviewer 4 Report

The manuscript by Jien et al. focuses on the extraction and microwave assisted hydrolysis of collagen to antioxidant peptides from sea cucumbers. The manuscript offers a good command of the English language, is in scope with the journal and is very well structured, showing an extensive work in terms of laboratorial procedures. Thus, there are some clarifications I would like the authors to reply to, pending the recommendation to publish this manuscript.

Abstract: There are too many acronyms in the abstract, which should be reduced.

Introduction: Although amino acids are widely known, their acronyms should be detailed when they first appear in the manuscript.

Introduction: There should be more information about the sea cucumber. Habits, habitat and further details about this echinoderm.

Line 76: “it aggregation”.

Line 92: “This results”.

Line 147: Assisted what?

Line 158: “th”.

Line 183 to 184: Wouldn’t this also influence in the same way the other fractions?

Line 197 to 198: ??? For all concentrations?

Line 200 to 202: ??? 50%??? Looking at figure 6a I would say it was about 70%

Line 205 to 207: Why at 1 mg/mL if the biggest difference was found at 2 mg/mL?

Line 210: No, the sharply increase does not explain why ABTS if better than DPPH.

Figure 6a: There is no statistical difference from 4.5 mg/mL.

If ABTS radical scavenging activity was better than DPPH, they why did the authors use DPPH to analyse the fraction’s activity in the gel filtration chromatography.

Figure 8b: There is no statistical difference between Fa3, Fa5 and Fa1. Please rewrite the section referring to this. Can the authors explain why the error or standard deviation bars are so big in this assay when they are strangely small in all the other ones?

Line 265: This is quite obvious and should be integrated into the next sentence.

Table 1: There is no standard error or standard deviation in this table, which is not acceptable.

Line 307: Body wall?

Line 310: Water to neutral?

Section 3.3: The different microwave power used should be clearly stated.

Section 3.4.1 to 3.4.2: Why was the ABTS left to react in the dark but the DPPH assay was left at room temperature? Light can oxidize both.

Section 3.8: Where there 3 different samples of 3 repetitions of the same sample? How were the results expressed, in standard error, standard deviation….? Why Duncan’s test?

Reviewer 5 Report

In my opinion the manuscript is interesting and suitable for publication in “Marine Drugs”. The language should be modified carefully. Some typing mistakes should be corrected.